# Reproductive tract and pouch anatomical variability across the reproductive phases in female common opossum (*Didelphis marsupialis* Linnaeus, 1758)

**Andrés Sepúlveda-Vásquez[1], Claudia P. Ceballos[2], Lynda J. Tamayo-Arango[1]***

**1** Facultad de Ciencias Agrarias, Grupo de investigación CIBAV, Escuela de Medicina Veterinaria, Universidad de Antioquia, Medellín, Colombia, **2** Facultad de Ciencias Agrarias, Grupo de investigación GAMMA, Escuela de Medicina Veterinaria, Universidad de Antioquia, Medellín, Colombia

* lynda.tamayo@udea.edu.co

## Abstract

Colombia's diverse ecosystems are home to various marsupial species known for its distinctive reproductive traits. Limited research has explored the reproductive anatomy of *Didelphis marsupialis*, particularly regarding variations associated with reproductive phase. This study aimed to characterize the reproductive anatomy of female *D. marsupialis* and assess its relationship with reproductive phases. We analyzed 57 female opossum cadavers using dissection, histology, and biometry. Specimens were classified by life stage based on dental chronology and by reproductive phase—interestrus, proliferative, and diestrus—based on ovarian features. Among the specimens, 79% lacked pouch young, while 21% carried young with an average of 3 individuals (range = 1–7). Uterine dimensions varied with the reproductive phase, with the pregnant female displaying the largest measurements. Five teats were commonly observed in adults (range = 0–10), while subadults exhibited the highest mean teat count (eleven), suggesting an adaptability to the reproductive demands. We identify six distinct anatomical variations in the vaginal complex, including the inconstant presence of a vaginal sinus septum and diverse *cul-de-sac* configurations. These variations allow us to reconstruct the temporary formation and subsequent regression of the birth canal, characterized by an invagination of the vaginal sinus and epithelial lining during pregnancy, followed by a postpartum involution. Additionally, we propose to use the term "urogenital canal" over "urogenital sinus" as it is more accurate anatomically. Future research should address the timing of the birth canal formation and regression and its relationship with the mammary gland development in living individuals.

**Data availability statement:** All relevant data are within the manuscript.

**Funding:** The author(s) received no specific funding for this work.

**Competing interests:** The authors have declared that no competing interests exist.

## Introduction

Colombia is renowned for its rich biodiversity, which has attracted the attention of researchers worldwide [1,2]. Among the many species that inhabit the country, marsupials stand out due to their unusual reproductive system and peculiar developmental process. Unlike most mammals, marsupial young are born in an extremely altricial state and complete their development within the mother's pouch. The common opossum (*Didelphis marsupialis*), one of six species in the *Didelphis* genus, is a nocturnal, nomadic marsupial that thrives across a range of environments in the Americas, including urban and peri-urban areas [3–6].

The reproductive system of marsupials consists of two ovaries, two uterine tubes, two uteri (each with its respective uterine body and cervix), and a vaginal complex that includes a vaginal sinus, a urogenital sinus, a birth canal (formed within the urogenital cord), two lateral vaginas, and a urogenital canal that opens into a common duct shared with the digestive system [7–9]. In marsupials, numerous anatomical variations occur both at both the interspecific and intraspecific levels, particularly within the vaginal complex. This structure may present a complete vaginal septum with two independent *cul-de-sacs* in females that have not yet given birth, and an incomplete vaginal septum with a single *cul-de-sac* in females that have previously given birth [7,10,11].

The reproductive anatomy of female opossums has been studied in species like *D. virginiana* [12] and *D. albiventris* [13], but these studies are limited by small sample sizes (n = 1–5) which makes difficult to assess intraspecific anatomical variations. Only one study evaluated the morphology of the female genital organs in *Didelphis* sp. [14], but no study has addressed the reproductive anatomy of *D. marsupialis*, and how it varies with the estrus cycle.

Understanding the reproductive anatomy of *D. marsupialis* is important because of several reasons. First, it would improve veterinary care and upgrade surgical techniques in Wildlife Rescue and Rehabilitation Centers (CAVs) in regions where human-opossum conflicts are common [15]. For example, in Antioquia, Colombia, common opossums are frequently injured by domestic animals, attacked by humans, electrocuted, or involved in traffic accidents [6]. Second, in several countries *D. marsupialis* is raised in captivity for human consumption such as Brazil, Colombia, and Trinidad and Tobago [16–18]. Thus, this knowledge could enhance their reproductive strategies, including the use of biotechnologies [19]. In a broader sense, understanding the reproductive system of the common opossum can add information on the marsupial adaptations to provide insight into the evolution of the mammalian reproductive systems [20]. This study aims to characterize the reproductive anatomy of female *D. marsupialis* in relation to reproductive phase by identifying anatomical variations and their associated factors.

## Methodology

### Ethical considerations

This research was approved by the Animal Experimentation Ethics Committee (CEEA) of Universidad de Antioquia, under Act 134 from August 2020.

## Specimens used

A total of 57 female opossum (*Didelphis marsupialis*) cadavers were received between 2018 and 2024 from regional environmental authorities, including the Metropolitan Area of Aburrá Valley (AMVA) and the Regional Autonomous Corporation of the Negro and Nare River Basins (CORNARE). Of the 57 specimens, 55 were stored at −20°C at the Animal Anatomy Laboratory of the Universidad de Antioquia until further use. The remaining 2 individuals, received less than 6 hours postmortem, were used for histological analysis of the ovaries and urogenital cord. Female specimens with intact reproductive tracts, irrespective of age or size, were included in the study, and those with evident lesions that hindered biometric measurements or reproductive tract injuries were excluded. Specimens used for macroscopic anatomical descriptions were preserved through soft embalming with a solution of ethanol, benzalkonium chloride, and propylene glycol, perfused via the common carotid artery [21]. Then an external examination was done to each female opossum to collect biometric data, determine the life stage, and inspect the marsupium and record information on the young if any.

## External examination

Biometric data collected included straight length from nose tip to tail tip, straight length from *nucha* to cloaca, straight length of the skull, and skull width. The life stage was determined based on the dentition. The absence of canines with erupting teeth indicated a juvenile stage (approximate age: 1 month); the presence of canines, two permanent premolars and a deciduous third premolar indicated subadult stage (approximate age: 7–10 months); and the full permanent dentition (I5/4, C1/1, Pm3/3, M4/4) indicated an adult stage (approximate age: > 10 months) [22,23].

The mammary gland was examined for signs of lactation such as an enlarged mammary gland or elongated teats. The number of teats was recorded, and they were classified as undeveloped (with a button-shaped), inactive (papillary shape but without turgid mammary gland), or active (papillary shape with turgid mammary gland). The number of pouch young was recorded if any, and their age was determined according to Flórez & Vivas (2020) [24]. They were classified as follows: less than one week old if no structural distinctions are observed, between one and two weeks old if ears remain adhered to the skull, and pelvic limb development is rudimentary, between two and three weeks old if sexual organs are visible, and three weeks or older if vibrissae and fully formed auricles are present (Fig 1).

## Dissection process

A longitudinal incision along the ventral midline, circumventing the pouch, was made to access the abdominal cavity. The pelvic symphysis was separated using a rib cutter to allow complete visualization of the reproductive tract. The ovaries and reproductive tract of each specimen were described, and the uterus were measured (length and width). Incisions

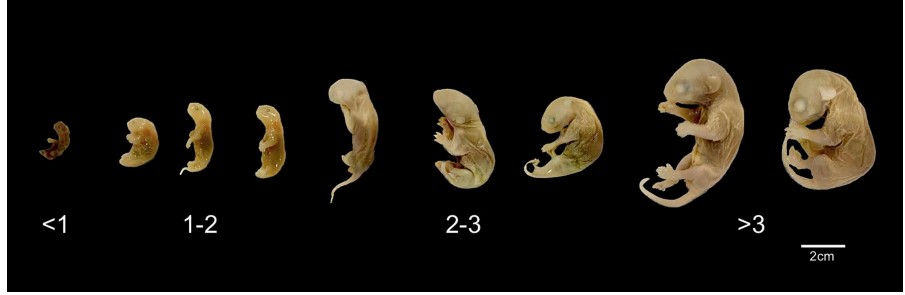

**Fig 1. Pouch young of *Didelphis marsupialis* classified by age.** Less than one week (<1), from one to two weeks (1-2), from two to three weeks (2-3), and more than three weeks (>3).

were made in the vaginal sinus and urogenital canal to examine the presence and morphology of the septum, vaginal *cul-de-sacs*, and the birth canal.

The phase of the reproductive cycle of each female was determined as interestrus, proliferative, and diestrus or luteal [8]. The interestrus phase was characterized by developing ovarian follicles without preovulatory turgid follicles or by the absence of ovarian structures. It could be a lactational anestrus, evidenced by mammary gland development with or without pouch young, or an anestrus due to an estrus cycle without mating (the latter being challenging to confirm in cadavers). In the proliferative phase was defined by the presence of preovulatory follicles, and evident mucosal hyperplasia in various segments of the reproductive tract. Finally, the diestrus is determined by the presence of *corpora lutea* in the ovary, or the presence of uterine fetuses (pregnancy).

Ten reproductive tracts were dissected *ex situ*, with transverse sections taken at the vaginal sinus, urogenital cord, and urogenital canal. These sections were examined under a stereoscope to confirm the presence of an intervaginal septum or a birth canal.

Finally, complete reproductive tracts from two specimens—an adult female with pouch young approximately four weeks old and a subadult opossum—were fixed in 10% buffered formalin for subsequent histological analysis. Specifically, we obtained fragments from the ovaries, uterine tubes, uterus, vaginal sinus, lateral vaginas, along with transverse sections from each third of the urogenital cord and birth canal for histological analyses. All tissue fragments were processed using a Thermo Scientific tissue processor (Thermo Excelsior AS500) for dehydration and clearing, followed by paraffin embedding. Sections were cut at 4 *µ*m using a microtome and stained with hematoxylin and eosin for optical microscopy analysis (Microscope Olympus BX53).

### Data analysis

Descriptive analyses summarized the main characteristics of the sample. Frequency tables were created for qualitative variables. Central tendency measures (mean, median, and mode) and dispersion measures (standard deviation and range) were calculated for quantitative variables to characterize biometric data.

## Results

### Age determination and biometry

In the determination of age through dental chronology, most individuals were classified as adults (91.2%, n = 52), a smaller number of individuals were categorized in the subadult stage (3.5%, n = 2) or in the juvenile stage (5.3%, n = 3). The morphometric analysis identified differences in various body measurements when comparing life stages: adult, subadult, and juvenile. (Table 1).

It was observed that 12 individuals (21%) had young in the pouch. For these females, variable combinations in the number and age of the young were observed, ranging from one to seven young, with ages spanning from less than one week to more than three weeks. Among the individuals without pouch young, one (1.75%) had 12 developing young in the uterus (pregnancy stage) (Fig 1).

**Table 1. Morphometry of common opossum females discriminated by life stage (cm ± SD).**

| Life stage | n | Skull length | Skull width | *Nucha*-cloaca length | Nose-tail length |
|---|---|---|---|---|---|
| Juvenile | 3 | 4.13 ± 0.06 | 2.03 ± 0.06 | 9.43 ± 0.55 | 23.80 ± 1.14 |
| Subadult | 2 | 8.35 ± 0.49 | 3.95 ± 0.35 | 24.75 ± 1.77 | 63.85 ± 4.74 |
| Adult | 52 | 10.56 ± 0.80 | 5.27 ± 0.62 | 32.30 ± 2.09 | 77.92 ± 5.65 |

## Morphology of the reproductive tract

The reproductive tract of the female common opossum is located in the hypogastric abdominal region and within the pelvic cavity, between the rectum and the bladder. Two epipubic fat pads were situated between the tract and the epipubic bones. The tract is composed of two ovaries, two uterine tubes, two uteri, and the vaginal complex (Fig 2).

## Ovaries and uterine tubes

The ovaries were located at the cranial end of each uterus and were associated with a suspensory ligament. This ligament was divided in two portions: cranially, it was closely related to the ureter in its cranial portion; and caudally, it fanned out with fibers extending towards the mesocolon, kidney, and ureter. The ovaries were encased in the ovarian bursa, formed ventrally by the interuterine ligament and dorsally by the broad uterine ligaments. It was observed that the mesosalpinx and meso-varium, which support the uterine tube and ovaries respectively, originated from the interuterine ligament (Fig 2). The uterine

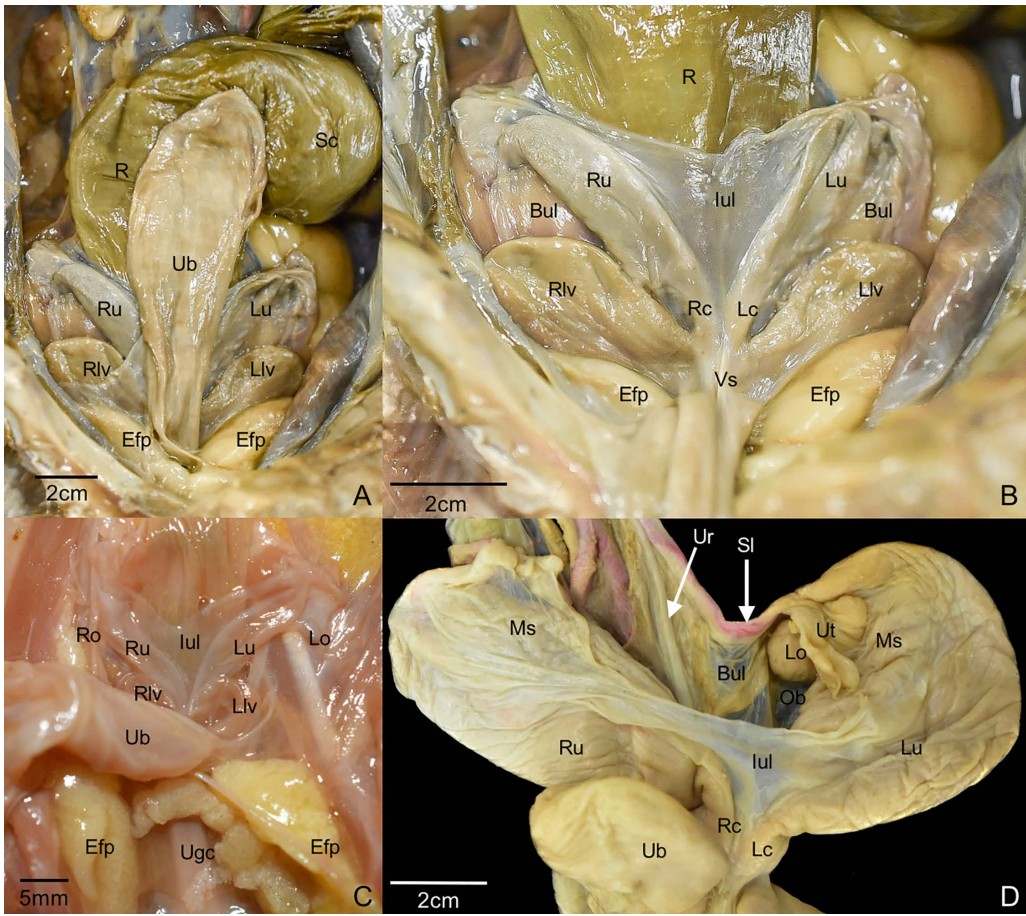

**Fig 2. Ventral view of the reproductive tract in situ, with the bladder in its anatomical position. (A)**, with the bladder retracted caudally in adult (B) and juvenile (C) specimens, and a detailed view of the left ovary and ovarian bursa *ex-situ* **(D)**. Bul, broad uterine ligament; Efp, epipubic fat pad; Lc, left cervix; Llv, left lateral vagina; Lo, left ovary; Lu, left uterus; Iul, interuterine ligament; Ms, mesosalpinx; Ob, Ovarian bursa; R, rectum; Rc, rigth cervix; Rlv, right lateral vagina; Ro, right ovary; Ru, right uterus; Sc, sigmoid colon; Sl, suspensory ligament of the ovary with ovarian vasculature; Ub, urinary bladder; Ugc, urogenital canal; Ur, ureter; Ut, uterine tube; Vs, vaginal sinus.

tube displayed an infundibulum, an ampulla, and an isthmus (Fig 3A), and was lined internally by simple ciliated columnar epithelium (Fig 3B).

From the individuals classified in interestrus phase (n = 44, 85.9%), most of them had ovaries with, apparently, no visible structures (follicles/corpora lutea) (n = 36, 71.9%), while the other females exhibited multiple small follicles (n = 8, 14%). Seven individuals were classified in diestrus phase (n = 7 12.2%), and 1 female was found pregnant (n = 1 1.7%). Juveniles and subadults had no structures in their ovaries and were classified as immature individuals (n = 5, 9%).

Histological analysis of the ovaries in adult opossums revealed follicles at all stages of development, a few primordial follicles, numerous atretic follicles, and abundant stromal tissue. Conversely, in subadult females, primordial follicles predominated, with few secondary, tertiary, and atretic follicles and sparse stromal tissue (Fig 4 and 5).

## Uterus

Each uterus consisted of a cervix (muscular portion) and a body (glandular portion), arranged obliquely in a 'V' shape, with their cranial ends directed laterally. An extensive interuterine ligament was observed medially. Laterally, the broad uterine ligament extended from the ovarian vasculature to the urogenital canal (Fig 2). Each uterus was connected to the vaginal sinus by its respective cervix, which had an external uterine opening directed slightly lateral (Fig 6A). Each cervix had a vaginal portion, i.e., a caudal projection forming a fornix (Fig 6B). As expected, gravid females exhibited the largest uterus in length and width, while immature individuals exhibited the smallest uterus. In addition, females in diestrus exhibited larger uterus, compared to individuals in interestrus (Table 2).

## Vaginal complex

The vaginal complex consisted of a vaginal sinus, two lateral vaginas, a urogenital cord, and a urogenital canal. The vaginal sinus (Figs 2B and 6) is the convergence point of the uteri in their external uterine openings with the lateral vaginas in their internal vaginal openings. The mucosa of the vaginal sinus exhibited several folds directed from the lateral vagina toward the cervix, along with smaller vaginal folds extending caudally toward the *cul-de-sac* (Fig 6B).

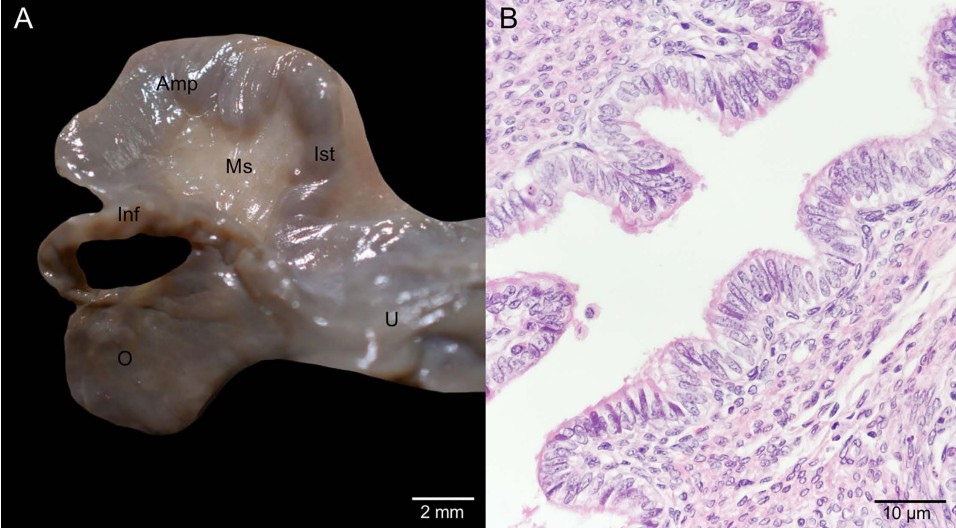

**Fig 3. Uterine tube from an adult specimen in interestrus. (A)** Macrophotograph showing the portions of the uterine tube. Amp, ampulla; Inf, infundibulum; Ist, isthmus; Ms, mesosalpinx; O, ovary; U, uterus. **(B)** Microphotograph of the ampulla displaying its epithelial lining.

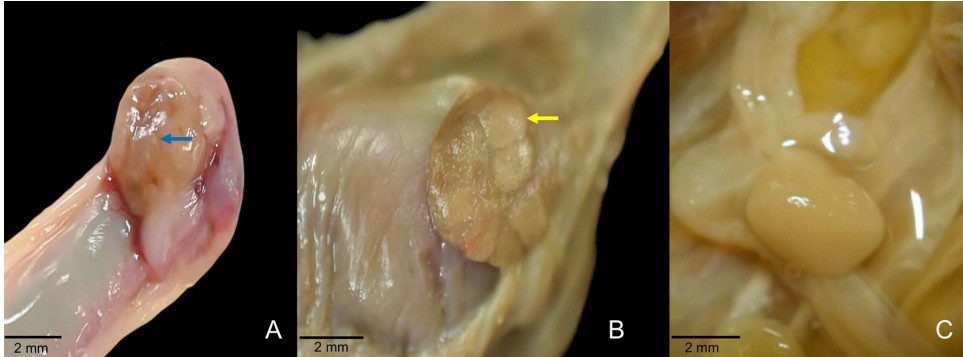

**Fig 4. Ovaries of *Didelphis marsupialis* with different ovarian structures. (A)** Ovary with multiple follicles (blue arrow) corresponding to a female in interestrus. **(B)** Ovary with *corpora lutea* (yellow arrow) of a female in diestrus. **(C)** Ovary without visible structures of a female in interestrus.

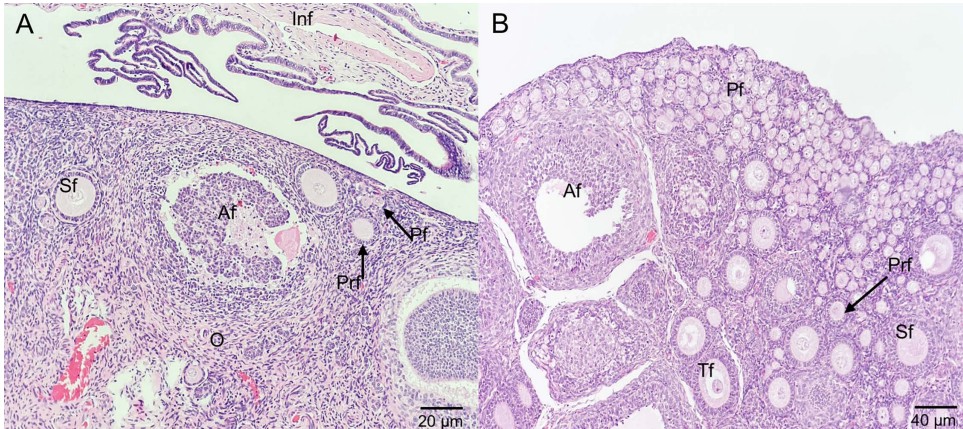

**Fig 5. Microphotographs of the ovaries of adult and subadult female opossums. (A)** Ovary and uterine tube of an adult. **(B)** Ovary of a subadult opossum. Af, atretic follicle; Inf, infundibulum; O, ovary; Pf, primordial follicles; Prf, primary follicle; Sf, secondary follicle; Tf, tertiary follicle.

In some females, the vaginal sinus had a midline septum dividing it into two independent compartments, also forming two caudal blind sacs referred to as *cul-de-sac* (Fig 6A y 6C). The septum was an inconstant structure, present in 19 of the 57 individuals studied, while the remaining 38 specimens exhibited a vaginal sinus without divisions, with the presence of a cranial fold in all the opossums, located dorsally, and a remnant caudal incomplete septum in some individuals, while in others there is no septum, with or without a *cul-de-sac* (Fig 7). The morphology of the *cul-de-sac* also varied: all individuals with a septum had a double and short *cul-de-sac*, whereas the other specimens lacking a septum could present variations in the *cul-de-sac* morphology: double and short with incomplete septum, double and long with incomplete septum, single and long without a septum, or single and short without a septum (Fig 7).

In the analysis of the relationship between *cul-de-sac* morphology and the reproductive phase of the females, the "double and short *cul-de-sac* with complete septum" category predominated with 19 individuals (33.3%), followed by 10 individuals with "double and short *cul-de-sac* with incomplete septum" (17.5%), mostly in the interestrus phase for both categories. The category "double and long with incomplete septum" was the least represented category with only 3 individuals in interestrus (5.3%). The "single and long without septum" category, with 12 individuals (21.1%), displayed the greatest diversity of reproductive phases, including one case of pregnancy. On the other hand, "no *cul-de-sac*",

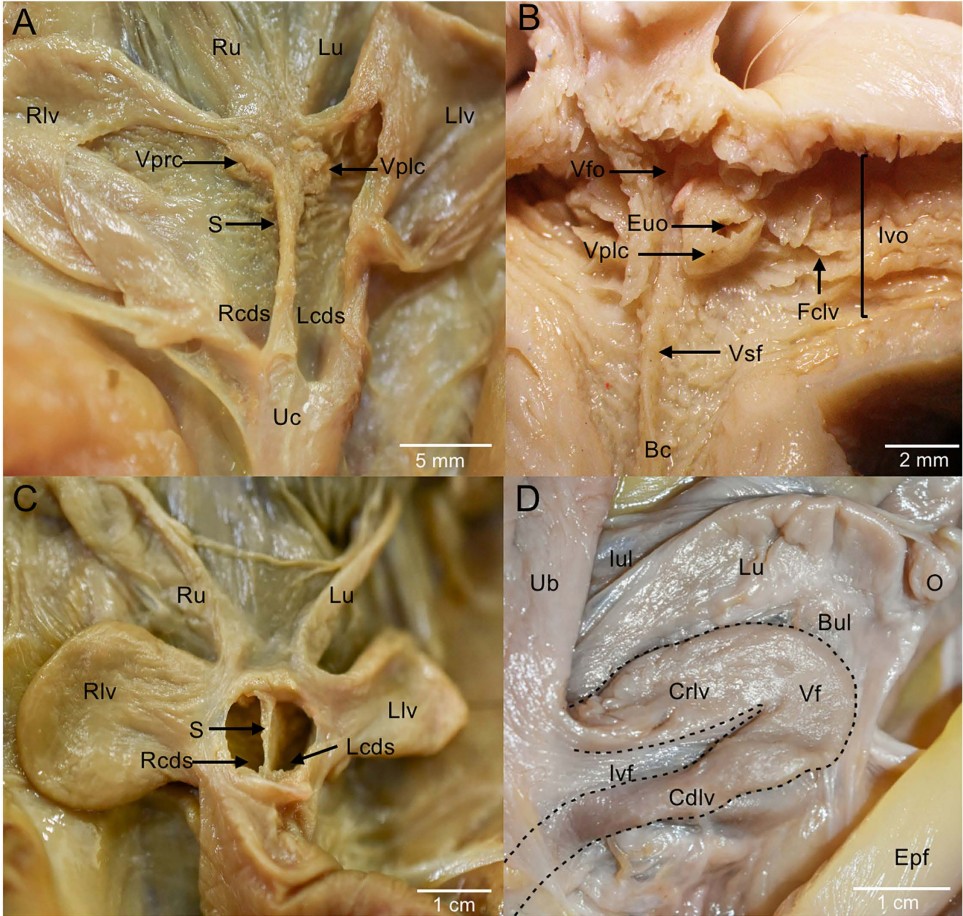

**Fig 6. Ventral view of the vaginal sinus. A.** Vaginal sinus with double *cul-de-sac*. **B.** Vaginal sinus with an open birth canal (no *cul-de-sac*). **C.** Ventral view of the vaginal sinus fully divided by the vaginal sinus septum. **D.** Ventral view of the left lateral vagina delimited with a dotted line. Bc, birth canal; Bul, broad uterine ligament; Crlv, cranial portion of the lateral vagina; Cdlv, caudal portion of the lateral vagina; Epf, epipubic fat pad; Euo, external uterine opening; Fclv, mucosal folds of the cranial lateral vagina; Iul, interuterine ligament; Ivf, intervaginal fold; Ivo, internal vaginal opening; Lcds, left *cul-de-sac*; Llv, left lateral vagina; Lu, left uterus; O, ovary; Rcds, right *cul-de-sac;* Rlv, right lateral vagina; Ru, right uterus; S, vaginal sinus septum; Ub, urinary bladder; Uc, urogenital cord; Vf, vaginal flexure; Vfo, vaginal fornix; Vplc, vaginal portion of the left cervix; Vprc, vaginal portion of the right cervix; Vsf, vaginal sinus fold.

**Table 2. Uterine measurements of *Didelphis marsupialis* in different reproductive phases (cm±SD).**

| Reproductive phase | n | Right uterus | | Left uterus | |
|---|---|---|---|---|---|
| | | Length | Width | Length | Width |
| Interestrus | 44 | 3.11±0.61 | 0.67±0.44 | 3.05±0.52 | 0.63±0.16 |
| Diestrus | 7 | 4.11±0.64 | 1.23±0.38 | 3.96±0.65 | 1.11±0.42 |
| Pregnancy | 1 | 5.10±NA | 1.80±NA | 5.30±NA | 2.00±NA |
| Proliferative | 0 | | | | |
| Immature* | 5 | 1.29±0.56 | 0.23±0.12 | 1.19±0.56 | 0.21±0.11 |

* Subadult and juvenile specimens.

representing 6 individuals with an open birth canal (10.5%), showed a balanced distribution between the diestrus and interestrus phases. The last category found, "single and short *cul-de-sac* without septum", was found in 7 individuals (12.3%), most of them in interestrus (Fig 7).

Out of the 19 (33.3%) females with complete septum, 8 (31.6%) of them had no pouch young, whereas only 1 female had pouch young (1.7%). From the females with incomplete septum or without septum (n = 38, 66.6%), 27 had no pouch young (47.4%), while 11 had young present (19.3%).

The lateral vaginas exhibited a U-shaped configuration, each with two distinct openings: an internal vaginal opening connecting to the vaginal sinus, and an external vaginal opening leading to the urogenital canal. Each lateral vagina consisted of two segments: the ascending vagina, which was thin, oriented cranially, and included both a fused section attached to the urogenital cord and a free portion oriented craniolaterally, followed by the descending vagina, which was thicker and ended at the vaginal sinus. An intervaginal fold was present between both portions of the lateral vaginas (Fig 6D).

The birth canal was found to be an inconstant structure. In some individuals, a duct in the longitudinal axis of the urogenital cord was identified, located between the lateral vaginas and dorsal to the urethra, connecting the vaginal sinus to the urogenital canal. In other cases, it was found closed, formed by connective tissue only, located between the vaginal sinus and the urogenital canal (Figs 8 and 9).

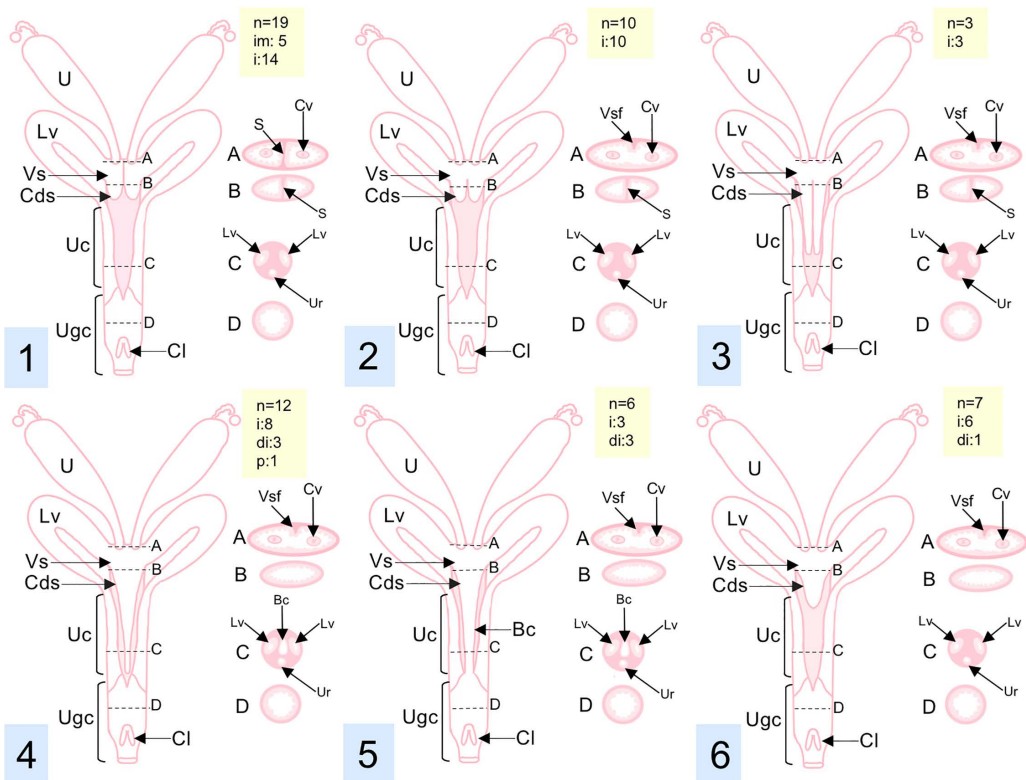

**Fig 7. Variations in the reproductive tract of *Didelphis marsupialis* females.1.** Double and short *cul-de-sac* with complete septum. 2. Double and short *cul-de-sac* with incomplete septum. 3. Double and long *cul-de-sac* with incomplete septum. 4. Single and long *cul-de-sac* without septum. 5. No *cul-de-sac (*open birth canal). 6. Single and short *cul-de-sac* without septum. Cross-section of the vaginal sinus **(A)**, *cul-de-sac* **(B)**, urogenital cord **(C)**, and urogenital canal **(D)**. Reproductive phase: Di, diestrus; I, interestrus, Im, immature; P, pregnancy. Structures: Bc, Birth canal; Cl, clitoris; Cds, *cul-de-sac*; Cv, cervix; Lv, lateral vagina; S, vaginal sinus septum; U, uterus; Ur, urethra; Uc, urogenital cord; Ugc, urogenital canal; Vs, vaginal sinus; Vsf, vaginal sinus fold. Abbreviations in yellow box: di, diestrus, i, interestrus, im, immature individuals, n, sample size, p, pregnant.

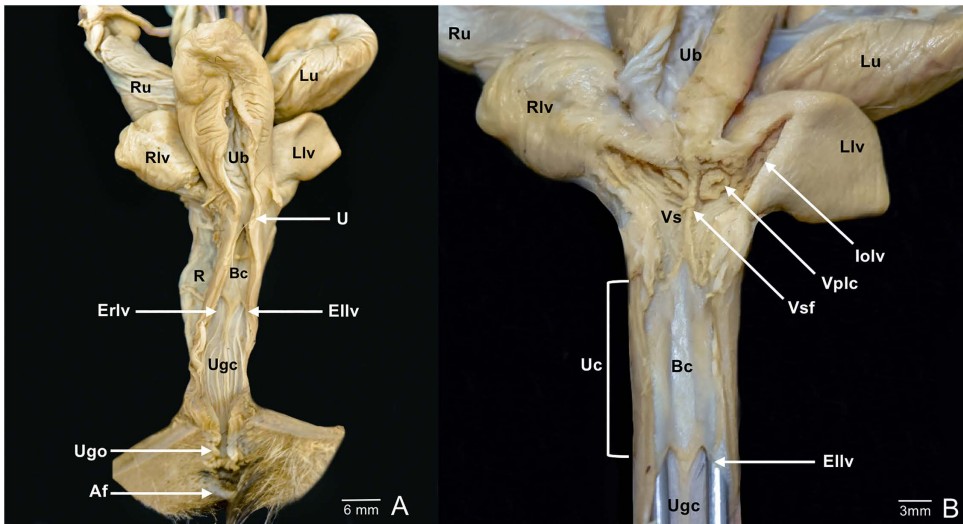

**Fig 8. Complete reproductive tract. A.** Ventral view with a longitudinal incision. **B.** Detail of the vaginal complex with an open birth canal; the ventral portion along with the urethra was removed; to visualize the lumen of the lateral vaginas, two rounded metal probes were inserted in a caudal-to-cranial direction. Af, anal fold; Bc, birth canal; Ellv, external opening of the left lateral vagina; Erlv, external opening of the right lateral vagina; Iolv, internal opening of the left lateral vagina; Llv, left lateral vagina; Lu, left uterus; R, rectum; Rlv, right lateral vagina; Ru, right uterus; U, urethra; Ub, urinary bladder; Uc, urogenital cord; Ugc, urogenital canal; Ugo, urogenital opening; Vplc, vaginal portion of the left cervix; Vs, vaginal sinus; Vsf, vaginal sinus fold.

Histological analysis of the urogenital cord in subadult opossums revealed a weak point lacking connective tissue at the site where the birth canal would form (Fig 9D). In an adult individual with pouch young >3 weeks old, the histology of the urogenital cord showed connective tissue in the caudal and middle thirds of the cord, while in the cranial third, a duct was observed with a muscular layer lined by simple columnar epithelium. This epithelium exhibited primary papillary formations, mild autolysis, and necrosis of individual cells, with apoptosis and mineralization foci. Few lymphocytes were present, without polymorphonuclear cells or marked inflammatory reactions (Fig 10).

The urogenital canal opened into a false cloaca through the urogenital opening and displayed longitudinal folds directed toward the external vaginal openings (Figs 8 and 11A). The clitoris was located ventrally in the most caudal portion of the urogenital canal, near the urogenital opening; it had a bifid glans oriented caudally and a groove extending from the bifurcation to the middle portion of the glans (Fig 11B). The broad uterine ligament also supported the lateral vaginas and the urogenital canal (Fig 7).

## The Pouch

The morphology of the pouch among juvenile, subadult, and adult individuals was highly variable in shape and number of teats present (Fig 12). In juveniles and subadults, a U-shaped skin fold was observed around button-shaped teats. In two juveniles, 9 teats were found arranged in two rows parallel to the ventral midline, each with 4 teats and a central teat slightly cranial to the third row of teats. In one juvenile and two adults, no teats could be identified, and smooth skin was found inside the skin fold (Fig 12D).

In the two subadults studied, 11 teats were observed arranged in two rows of 5 teats parallel to the ventral midline, with a central teat slightly cranial to the fourth row of teats. In adults, a well-developed skin fold was observed, more developed in the caudal part, fully covering the teats and leaving only a small cranial opening. Regarding the number and arrangement of teats, significant variability was found, ranging from 0 to 11 teats, which could be button-shaped, papillary-shaped, or fully elongated depending on their physiological state. Some individuals presented teats in all three stages.

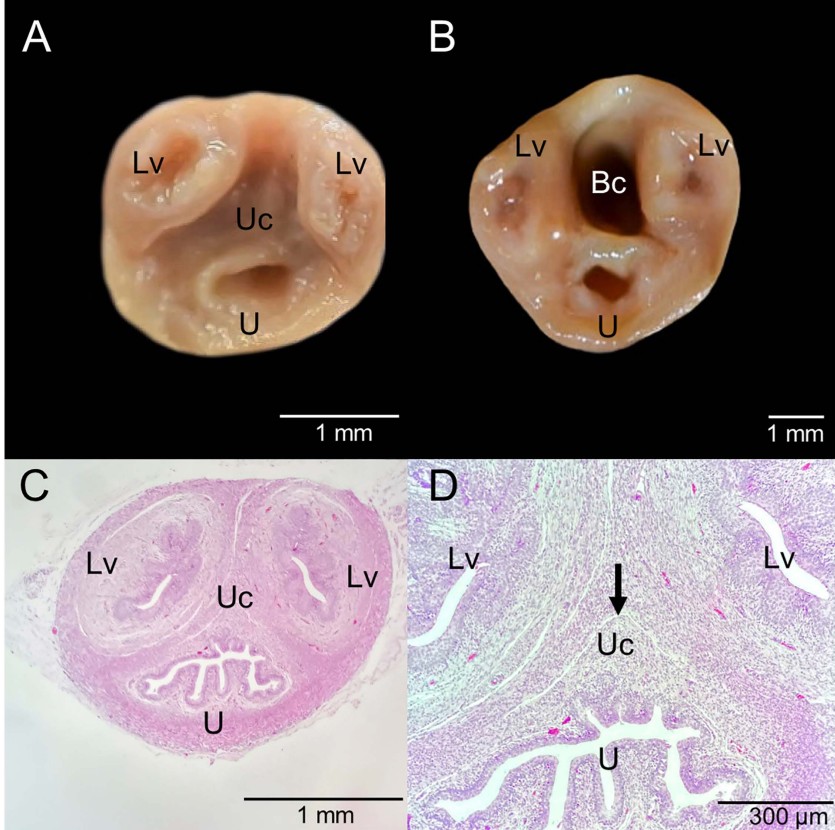

**Fig 9. Urogenital cord (As an anatomical reference, the urethra runs ventrally). A.** Cross-section without a formed birth canal. **B.** Cross-section with a formed birth canal. **C.** Histological section with a closed birth canal. **D.** Close-up of the histological section with a closed birth canal. Bc, birth canal; Lv, lateral vagina; U, urethra; Uc, urogenital cord. Point of birth canal formation (black arrow).

The most common arrangement found was 5 teats located in two rows of two teats, where the cranial teats were farther apart than the caudal ones, and the fifth teat was located on the midline towards the cranial side (Fig 12A). This distribution was found in 23 cases, representing 44.2% out of the 52 adults evaluated. As for the total number of teats, subadult females showed a higher value, with a mean of 11 teats (± 0.00), compared to adults (5.48±2.16) and juveniles (6±5.20). Regarding inactive teats, the means were low across all groups. Adult females presented a mean of 1.08 (± 1.97), while both subadults and juveniles had no inactive teats. Active teats were more numerous in adult females, with a mean of 3.12 (± 2.38), whereas neither subadults nor juveniles showed active teats. Finally, undeveloped teats were more common in subadult and juvenile females, with means of 11±0 and 6±5.2, respectively. In contrast, adult females had a considerably lower mean of undeveloped teats (1.29±2.26).

## Discussion

In the present study we examined a large and variable sample spanning different life and reproductive phases, observing that the female reproductive tract of *Didelphis marsupialis* exhibited a configuration similar to that described in other marsupials. This configuration is characterized by a dual reproductive system, consisting of two ovaries, two uteri with independent cervices, a vaginal sinus, two lateral vaginas, a urogenital cord, and a urogenital canal [7–11,15].

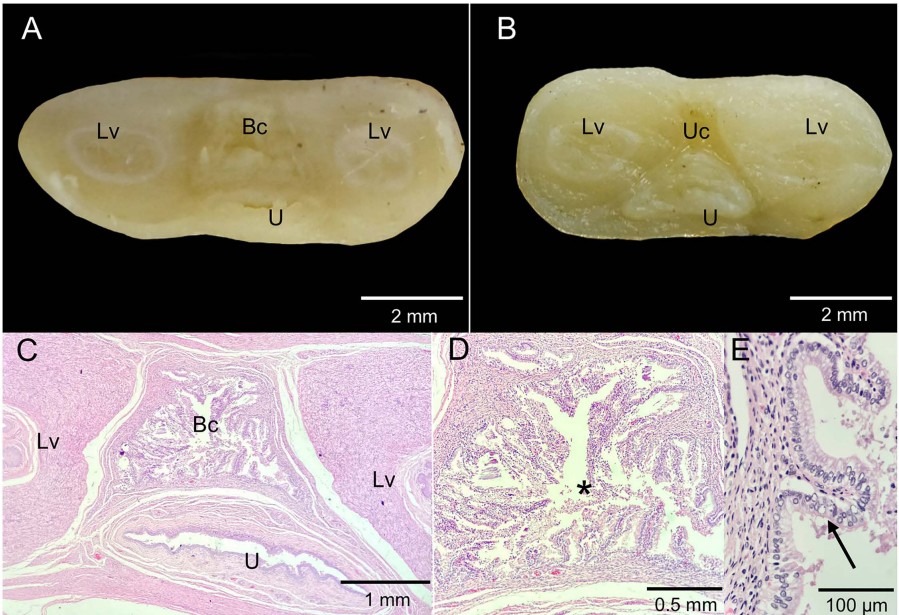

**Fig 10. Birth canal regression. A.** Cross-section of the urogenital cord at the cranial third. **B.** Cross-section of the urogenital cord in the middle third. **C.** Histological section of the urogenital cord at the cranial third. **D.** Birth canal in regression. **E.** Detail of the regressing birth canal displaying its mucosa at higher magnification. Bc, birth canal; Lv, lateral vagina; U, urethra; Uc, urogenital cord. Regressing birth canal (*), epithelium with apoptotic cells (Black arrow).

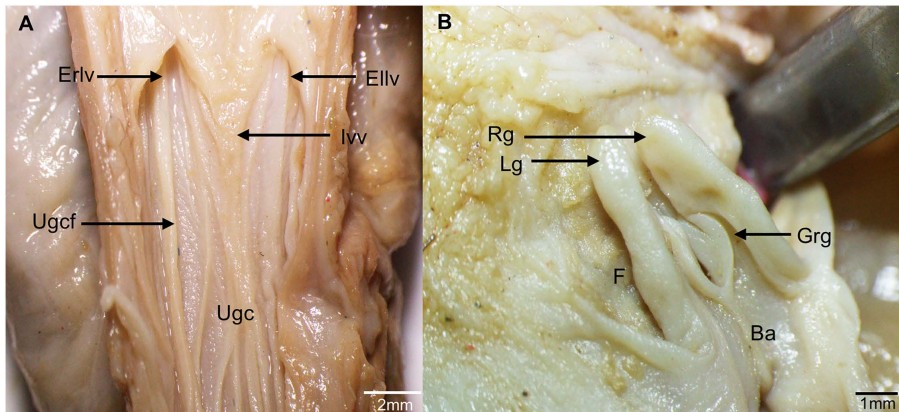

**Fig 11. Urogenital canal. A.** Ventral view. **B.** Detail of the clitoris. Ba, Base of the glans, oriented cranially; Ellv, external opening of the left lateral vagina; Erlv, external opening of the right lateral vagina; F, glans fossa; Grg, groove of the right portion of the glans; Ivv, intervaginal veil; Lg, left portion of the glans; Rg, right portion of the glans; Ugc, Urogenital canal; Ugcf, mucosal folds of the urogenital canal.

We classified six different anatomical variations of the reproductive tract, related to the reproductive phase. In agreement with our findings, previous studies have described different morphological configurations of the tract, reporting the presence of a vaginal septum in non-parous individuals, while this structure is absent in parous females. Additionally, the formation of a temporary birth canal has been observed in individuals close to parturition [8,15]. In macropodids, however, this birth canal persists as a permanent median vagina [11]. Based on the cadaveric examination, we suggest that the

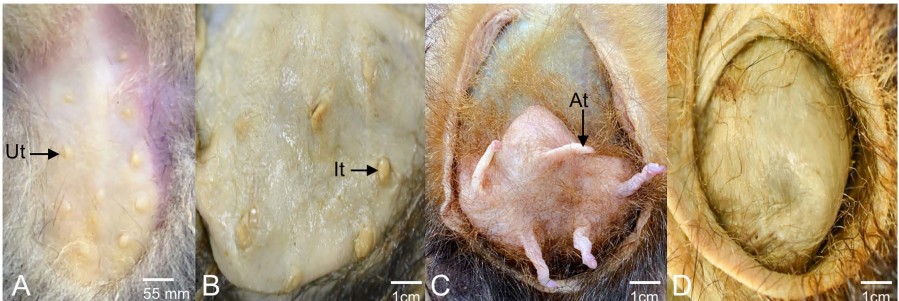

**Fig 12. Marsupium. A.** Subadult with 11 undeveloped teats. **B.** Pregnant with 8 inactive teats **C.** Adult with pouch young older than 3 weeks and 5 active teats. **D.** Adult without teats. At, active teat; It, inactive teat; Ut, undeveloped teat.

morphology of the *cul-de-sac* in *D. marsupialis* varies with the reproductive phase, becoming uniquely enlarged during pregnancy to form the birth canal at the time of parturition. Subsequently, within the first postpartum week, it shortens and divides again.

Our results regarding the morphology of the *cul-de-sac* and the presence of the birth canal, may indicate that the vaginal *cul-de-sac* unifies and elongates from cranial to caudal during gestation and, once birth occurs, it begins its involution from caudal to cranial, similar to what has been reported in koalas (*Phascolarctos cinereus*) [8] and common brushtail possum (*Trichosurus vulpecula)* [9].

The birth canal forms shortly before parturition and begins to regress within the first postpartum week [9,25,26], which is consistent with our findings. However, as this study was conducted on cadaveric specimens, further research involving live animals with close documentation is necessary to confirm the exact timing of birth canal formation and regression.

In marsupials, the urogenital sinus (segment from the urethral meatus to the external orifice of the tract) is elongated and forms the urogenital canal [7,27]. We propose that there is no distinction between the urogenital sinus and the urogenital canal, previously made for *D. albiventris* [13]. This terminology also contrasts with what has been reported in *D. virginiana* [12,28], where a urogenital cord was not described, and the urogenital sinus was referred to as a tubular structure into which the lateral vaginas and urethra open. We hope that this elucidation in the terminology could help prevent confusion in the anatomical description in marsupials.

The ovaries of *D. marsupialis* were enclosed within an ovarian bursa formed by the interuterine ligament and the broad ligament of the uterus, consistent with previous descriptions in *Didelphis sp.* [14] and other marsupials such as *Dendrolagus* [10], but contrasting with other marsupial species, such as *Petaurus breviceps*, which lack an ovarian bursa [15]. Regarding the position of the ovaries, in *Petaurus breviceps* they are positioned dorsolateral, like what we found in *D. marsupialis* [15].

Regarding the oviduct, marsupials are reported to have a wider isthmus compared to eutherians, which may play a role in sperm storage [29]. In our study, we found that *D. marsupialis* did not conform to this pattern, as the ampulla was the widest part of the tube, resembling the configuration found in eutherian mammals. This observation differs from previous reports on other marsupial species [15,26,29]. Nevertheless, this characteristic must be confirmed by morphoquantitative analysis of this structure.

Unlike Eutherians, the uterus in marsupials is composed only of the body and cervix. The uterine body is primarily fusiform; however, its shape and length can vary among species. For instance, in thylacine (*Thylacinus cynocephalus*), the uterine body was unusually elongated, while in Tasmanian devil (*Sarcophilus satanicus*), it was flattened. Additionally, the cervix of quolls (*Dasyurus*) was particularly long [7,26,30]. In *D. albiventris* [13] the uteri were reported to have cervix,

body, and horns, while we found that the uteri only presented body (corresponding to the glandular portion of the uterus) and cervix, consistent with what has been reported for marsupials [31].

The anatomical variations in the vaginal sinus septum are consistent with previous descriptions in *D. marsupialis* [31]. A complete septum was present in non-parous females (immature individuals), which separated the left and right portions of the genital tract. The incomplete septum was found in adults at different reproductive phases, suggesting that the septum is perforated after the first parturition, allowing communication between both sides of the vaginal sinus [32]. Our findings in *D. marsupialis* and reports on *Petaurus breviceps* confirm this pattern [15].

In some species, such as Matschie's tree-kangaroo (*Dendrolagus matschiei*), the division of the vaginal sinus is only partial, as the fold gradually decreases in height in the caudal direction until it disappears, remaining attached solely to the ventral wall and dividing only the cranial portion of the sinus [10]. In contrast, in *D. marsupialis*, the vaginal sinus displays multiple morphological presentations, with or without *cul-de-sacs.*

The lateral vaginas in marsupials are analogous to the vagina in eutherian mammals. However, while in eutherians this structure directly connects to the uterus, in marsupials, the lack of a direct connection between the urogenital canal and the uteri makes the lateral vaginas the only viable pathway for sperm transport in non-parous and most parous females [32]. In contrast, female macropods that have given birth possess a permanent median vagina, which allows direct access from the urogenital sinus to the cervices [19].

The morphology of the lateral vaginae is variable, being U-shaped and elongated in most marsupials [7,8,15,26,30], which agrees with our findings. In contrast, grey-bellied shrew opossum (*Caenolestes obscurus*) has lateral vaginae that were long and convoluted, while in *Trichosurus vulpecula* and honey possum (*Tarsipes rostratus*), they were short and straight. Additionally, in some species, the cranial portion of the lateral vagina forms a chamber that functions as a seminal receptacle [31]. We also found in *D. marsupialis* a thicker cranial portion of the lateral vagina. If it is a seminal receptacle, it must be studied in the future.

Near parturition, a birth canal forms between the lateral vaginas, connecting the vaginal sinus and passing through the midline via the urogenital cord [33–35]. In *D. marsupialis*, we identified a weak point in the midline of the urogenital cord composed of loose connective tissue, where the birth canal may develop. This feature has also been previously reported in *Phascolarctos cinereus* [8].

Our finding of a birth canal lined by a simple columnar epithelium in *D. marsupialis,* aligns with previous reports for *D. virginiana* [28] but differs from what has been described in *Phascolarctos cinereus*, where the birth canal is formed by vacuoles and lacks an epithelial lining [8]. The remnants of epithelial lining in the involuting birth canal observed in a female with pouch young older than three weeks suggests that the birth canal is formed by the invagination of the vaginal sinus along with its epithelium, consistent with reports for *Trichosurus vulpecula* [9]. These findings suggest that the presence of epithelium in the birth canal may represent an interspecific variation among marsupials.

The length of the birth canal varies among marsupial species. In *Macropodidae*, as well as in *D. marsupialis*, the vaginal sinus is separated from the urogenital canal by a long urogenital cord composed by connective tissue, which is equivalent in length to the lateral vaginas [36]. In contrast, other marsupial families, such as *Phalangeridae*, *Dasyuridae*, and *Vombatidae*, exhibit a closer relationship between the vaginal sinus and the urogenital canal, with only a thin layer of connective tissue separating them, which disappear in recently parous individuals [7]. In contrast with the findings for *Didelphis spp.* [14], a pseudovagina was not identified as such in *D. marsupialis*, although a unique vaginal sinus, vaginal *cul-de-sacs*, and birth canal could represent a functional equivalent to the structures described in *Didelphis spp*.

In the caudal portion of the urogenital canal of *D. marsupialis*, we identified a bifurcated clitoris with medial grooves, resembling the penis of this species. Similarly, other marsupial species with a bifurcated glans penis in males also exhibit a bifurcated clitoris in females [26,30,36]. In contrast, macropodids lack this characteristic and instead have an unforked clitoris [10]. Furthermore, for *Didelphis spp.* it is reported that the ventral wall of the urogenital sinus, near its distal end, contains erectile tissue, which forms a genital tubercle resembling a clitoris [14].

Although most marsupials possess a cloaca [37], didelphids have a urogenital orifice separated from the anus, as observed in *D. albiventris*, *Chironectes minimus*, and *Monodelphis* [13,27,38]. In *D. marsupialis*, we identified this characteristic organization typical of Didelphids, along with a dorsal fold of the anus that partially covers both openings, creating a pseudocloaca, as previously described [37].

Unlike the type 5 marsupium in sugar glider (*Petaurus breviceps*) (skin fold that completely covers the mammary area and have a clearly defined cranial opening), *D. marsupialis* has a type 3 marsupium (skin fold circularly covers the mammary gland with a central opening) [39]. Additionally, the teats exhibited more diverse anatomical configurations. While the *Petaurus breviceps* typically has four teats in two lateral pockets within the marsupium, *D. marsupialis* exhibited between 0 and 11 teats arranged in two rows parallel to the ventral midline. This arrangement varies according to the age and the reproductive phase, with teats at different phases of development (bud, papillary, or elongated), suggesting adaptability to the reproductive conditions of the individuals. This type of variability in the number and organization of teats differs not only from the *Petaurus breviceps* but also from other marsupial species, which tend to present a more stable marsupium structure [3, 40, 41].

Our results regarding the morphology of the teats, may suggest that the papillary form represents the resting state, awaiting the arrival of young, and subsequently enlarges once the offspring are born. The bud form of the teats appears to be vestigial or an inactive state that may become functional when the number of young requires it. Further studies on live animals are needed to confirm this hypothesis.

## Conclusions

The present study characterized the morphology of the reproductive tract and marsupium of *Didelphis marsupialis*. Anatomical variations of the reproductive tract were not related to body size in adult individuals but were mainly determined by the individual's reproductive phase. The variations in the birth canal presence highlight the dynamic formation and involution of this structure in response to the reproductive phase. Likewise, the number of teats appeared to be linked to reproductive needs and ontogenetic development. Future research should address the timing of the birth canal formation and regression, and its relationship with the mammary gland development including the number and development of teats. These efforts will facilitate comparative studies across species and enhance our understanding of marsupial reproductive biology.

## Acknowledgments

We thank the environmental authorities CORNARE and AMVA for providing the specimens. We are also grateful to J. Muñoz for his guidance in histological slide processing and analysis and to J. Badel for his constructive review and insightful comments on the manuscript. CIBAV thanks to the Strategy of Consolidation of Research Groups CODI 2023–2025, University of Antioquia, Medellín, Colombia.

## Author contributions

**Conceptualization:** Lynda Jhailú Tamayo-Arango, Andrés Sepúlveda-Vásquez.

**Data curation:** Lynda Jhailú Tamayo-Arango, Andrés Sepúlveda-Vásquez.

**Formal analysis:** Lynda Jhailú Tamayo-Arango, Andrés Sepúlveda-Vásquez.

**Investigation:** Lynda Jhailú Tamayo-Arango, Andrés Sepúlveda-Vásquez.

**Methodology:** Lynda Jhailú Tamayo-Arango, Andrés Sepúlveda-Vásquez, Claudia P. Ceballos.

**Project administration:** Lynda Jhailú Tamayo-Arango.

**Resources:** Lynda Jhailú Tamayo-Arango, Andrés Sepúlveda-Vásquez, Claudia P. Ceballos.



**Supervision:** Lynda Jhailú Tamayo-Arango.

**Validation:** Lynda Jhailú Tamayo-Arango, Andrés Sepúlveda-Vásquez, Claudia P. Ceballos.

**Writing – original draft:** Lynda Jhailú Tamayo-Arango, Andrés Sepúlveda-Vásquez.

**Writing – review & editing:** Lynda Jhailú Tamayo-Arango, Andrés Sepúlveda-Vásquez, Claudia P. Ceballos.

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
