## [Decision Letter · Decision Letter 0]

12 Aug 2025

Dear Dr. Tamayo-Arango,

Thank you for submitting your manuscript to PLOS ONE. After careful consideration, we feel that it has merit but does not fully meet PLOS ONE’s publication criteria as it currently stands. Therefore, we invite you to submit a revised version of the manuscript that addresses the points raised during the review process.

I now have expert reviews of this work. Although the reviewers found merit in the work there were many methodological and interpretation concerns that require a major revision with re-review.

We look forward to receiving your revised manuscript.

Kind regards,

James J Cray Jr., Ph.D.

Academic Editor

PLOS ONE

Journal Requirements:

2. We note that your Data Availability Statement is currently as follows: All relevant data are within the manuscript and in Supporting Information files.

4. Please include a copy of Table 3 which you refer to in your text on page 18.

Reviewers' comments:

Reviewer's Responses to Questions

**Comments to the Author**

1. Is the manuscript technically sound, and do the data support the conclusions?

Reviewer #1: Yes

Reviewer #2: Yes

2. Has the statistical analysis been performed appropriately and rigorously?

Reviewer #1: N/A

Reviewer #2: Yes

3. Have the authors made all data underlying the findings in their manuscript fully available?

Reviewer #1: Yes

Reviewer #2: No

4. Is the manuscript presented in an intelligible fashion and written in standard English?

Reviewer #1: Yes

Reviewer #2: Yes

Reviewer #1: Reproductive Tract and Pouch Anatomical Variability Across the Reproductive Phases in female common opossum (Didelphis marsupialis Linnaeus, 1758)

The authors present a comprehensive, coherent, and well-written study on the female reproductive system of Didelphis marsupialis. A considerable number of female specimens at different stages of the reproductive cycle were used to carry out this study; however, the number of pregnant females was limited, and no individuals in estrus were included. This may be considered its most significant limitation.

Aside from the point previously mentioned and the absence of Table 3, no major flaws are observed in the study’s design, development, results, discussion, or conclusions. This is a well-documented study, featuring high-quality figures and diagrams, and supported by appropriate references in both quantity and relevance. Therefore, the comments provided below may be considered minor and are intended to help improve the manuscript.

The manuscript revision in Word format has been uploaded to the system.

Reviewer #2: Introduction:

1. I disagree that the term "unique" is the best way to define the developmental process of marsupials. I think "peculiar" sounds better.

2. Marsupials are not kept in captivity for human consumption in Brazil. This is not a cultural practice in the country. Furthermore, I disagree that this is a justification for this study.

3. At some points, the text sounds like automatically translated.

Methodology

4. Since the methodology involves the use of cadavers collected by environmental agencies, it is important to mention how long after the animals' deaths they are still used for analysis. The practice of using dead animals for histological and anatomical analyses, even if "well-preserved," is controversial.

5. What bibliographic reference was used in the analysis and classification of the teats?

6. The author repeatedly cites "absence of ovarian structures" as a characteristic of immature ovaries. However, it is necessary to specify which structures are being referred to (probably those related to the ovarian cycle: ovarian follicles and corpus luteum). The ovaries have covering and supporting structures, for example, that exist independently of the ovarian cycle.

7. Only two specimens (adult and subadult) were selected for histological analyses of the complete reproductive tract. In addition to the very small n, other life stages and phases of the reproductive cycle were not considered. How is this decision justified? The article only shows the histology of some organs, but not the complete reproductive tract, as described in the methodology.

Results

8. Can the presence of young in the pouch be correlated with the time of year the animals were collected? Would these correlations apply to the number of young and the stage at which they were collected?

9. Does the topic "reproductive tract morphology" refer only to adult animals? If so, this needs to be clearly stated, since we have several developmental variables relevant to the context of the study.

10. Figure 3: Which segment of the fallopian tube is shown? From what life stage? In what reproductive phase?

11. Figure 2: Why weren't images for the "subadult" stage included?

12. Image A shows the infundibulum above and the ovary in the rest of the field. Highlighting the infundibulum in this image is unnecessary, especially since we don't have a corresponding image in the subadult animal (B). Comparing adult and subadult animals and showing the infundibulum in only one case creates confusion.

Removing the infundibulum from the field in image A would also make it unnecessary to label the "ovary" with the letter "O," as is currently done. The identification occurs in the cortical region of the ovary and does not encompass the entire organ fragment.

13. Figure 6: Which stage is shown?

14. Figure 9/10: Which stage is shown? 15. Are only the adult and subadult stages being shown?

Discussion

16. Throughout the text, it becomes unclear which life phase or reproductive cycle the author is referring to. Since there are so many analyses and so much information, it is necessary to specify which phase or stage is being discussed.

17. Some sections of the discussion are very similar to the results and sounds repetitive. This is particularly true in sections where the findings are listed again before being discussed.

18. The discussion highlights the questionable methodology used with cadavers.

19. The discussion compares the size of the isthmus and the ampulla of the uterine tube. However, no comparative histological analyses of the two segments were performed. The image of the uterine tube does not specify its specific segment or the animal's life/reproductive stage.

20. The discussion of the eithelium of the birth canal could be further explored. The author comments that "there are differences between species," and this is notable. However, what functional and evolutionary aspects might be relevant in this context?

21. Was histological analysis of the breasts considered?

22. Was analysis of the breasts considered when there were youngs at different stages of the pouch?

**Do you want your identity to be public for this peer review?** For information about this choice, including consent withdrawal, please see our Privacy Policy

Reviewer #1: **Yes: ** Matilde Lombardero

Reviewer #2: No

---

## [Author Response · Author response to Decision Letter 1]

19 Aug 2025

All the responses are in the attached file: Response to reviewers.

---

## [Decision Letter · Decision Letter 1]

22 Sep 2025

Reproductive Tract and Pouch Anatomical Variability Across the Reproductive Phases in female common opossum (Didelphis marsupialis Linnaeus, 1758)

PONE-D-25-30424R1

Dear Dr. Tamayo-Arango,

We’re pleased to inform you that your manuscript has been judged scientifically suitable for publication and will be formally accepted for publication once it meets all outstanding technical requirements.

Kind regards,

James J Cray Jr., Ph.D.

Academic Editor

PLOS ONE

Additional Editor Comments (optional):

Reviewer #1:

Reviewers' comments:

Reviewer's Responses to Questions

**Comments to the Author**

Reviewer #1: All comments have been addressed

2. Is the manuscript technically sound, and do the data support the conclusions?

Reviewer #1: Yes

3. Has the statistical analysis been performed appropriately and rigorously?

Reviewer #1: N/A

4. Have the authors made all data underlying the findings in their manuscript fully available?

Reviewer #1: Yes

5. Is the manuscript presented in an intelligible fashion and written in standard English?

Reviewer #1: Yes

Reviewer #1: The authors have adequately addressed all the issues raised. Only two minor comments remain to be noted:

-L240: figure 4 legend: it should say ‘corpora lutea’ instead of ‘corpora luteum’.

-Regarding Figure 7 quality: enhancing the contrast in this figure can be achieved as easily as using an image editing program such as Photoshop: by navigating to Image > Adjustments > Curves and modifying the green channel to intensify the pink tones with minimal impact on the other colors.

An example is provided.

**Do you want your identity to be public for this peer review?** For information about this choice, including consent withdrawal, please see our Privacy Policy

Reviewer #1: **Yes: ** Matilde Lombardero

---

## [Editor Report · Acceptance letter]

PONE-D-25-30424R1

PLOS ONE

Dear Dr. Tamayo-Arango,

I'm pleased to inform you that your manuscript has been deemed suitable for publication in PLOS ONE. Congratulations! Your manuscript is now being handed over to our production team.

Kind regards,

on behalf of

Dr. James J Cray Jr.

Academic Editor

PLOS ONE